# Simulation of spontaneous G protein activation reveals a new intermediate driving GDP unbinding

Xianqiang Sun[1†], Sukrit Singh[1†], Kendall J Blumer[2], Gregory R Bowman[1,3]*

[1]Department of Biochemistry and Molecular Biophysics, Washington University School of Medicine, Missouri, United States; [2]Department of Cell Biology and Physiology, Washington University School of Medicine, Missouri, United States; [3]Center for Biological Systems Engineering, Washington University School of Medicine, Missouri, United States

**Abstract** Activation of heterotrimeric G proteins is a key step in many signaling cascades. However, a complete mechanism for this process, which requires allosteric communication between binding sites that are ~30 Å apart, remains elusive. We construct an atomically detailed model of G protein activation by combining three powerful computational methods: metadynamics, Markov state models (MSMs), and CARDS analysis of correlated motions. We uncover a mechanism that is consistent with a wide variety of structural and biochemical data. Surprisingly, the rate-limiting step for GDP release correlates with tilting rather than translation of the GPCR-binding helix 5. β-Strands 1 – 3 and helix 1 emerge as hubs in the allosteric network that links conformational changes in the GPCR-binding site to disordering of the distal nucleotide-binding site and consequent GDP release. Our approach and insights provide foundations for understanding disease-implicated G protein mutants, illuminating slow events in allosteric networks, and examining unbinding processes with slow off-rates.
DOI: https://doi.org/10.7554/eLife.38465.001

*For correspondence:
g.bowman@wustl.edu

†These authors contributed equally to this work

Competing interests: The authors declare that no competing interests exist.

## Introduction

Heterotrimeric G proteins are molecular switches that play pivotal roles in signaling processes from vision to olfaction and neurotransmission (*Oldham and Hamm, 2006*; *Oldham and Hamm, 2008*; *Johnston and Siderovski, 2007*). By default, a G protein adopts an inactive state in which guanosine diphosphate (GDP) binds between the Ras-like and helical domains of the α-subunit (Gα, *Figure 1*). A dimer consisting of the β- and γ-subunits (Gβγ) also binds Gα. G protein-coupled receptors (GPCRs) trigger G-protein activation by binding Gα and stimulating GDP release, followed by GTP binding to Gα and dissociation of Gα from Gβγ. Gα and Gβγ then interact with effectors that trigger downstream cellular responses. Gα returns to the inactive state by hydrolyzing GTP to GDP and rebinding Gβγ. Given the central role Gα plays, a common Gα numbering scheme (CGN) has been established to facilitate discussion of the activation mechanisms of different Gα homologs (*Flock et al., 2015*). For example, the notation Lys52[G.H1.1] indicates that Lys52 is the first residue in helix 1 (H1) of the Ras-like domain (also called the GTPase domain, or G). S6 refers to β-strand 6 and s6h5 refers to the loop between S6 and H5.

Strikingly, the GPCR- and nucleotide-binding sites of Gα are ~30 Å apart (*Figure 1*), but the allosteric mechanism coupling these sites to evoke GDP release remains incompletely understood (*Oldham and Hamm, 2008*). Biochemical and structural studies have elucidated some key steps, but the entire process has yet to be described in atomic detail. Early studies of Gα subunits revealed structures of the GDP- and GTP-bound states, as well as the transition state for GTP hydrolysis

**eLife digest** Cells communicate with each other by exchanging chemical signals, which allow them to coordinate their activities and relay important information about their environment. Often, cells secrete specific signals into their surroundings, which are then picked up by a receiving cell that has the right receptors to recognize the message. Once the signal attaches to the receptor, its shape or activity changes, which in turn triggers cascades inside the cell to convey the signal, much like a circuit would.

A group of proteins called heterotrimeric G-proteins play an important role in these pathways. They act as molecular switches inside the cells to help transmit signals from the outside of the cell to the inside. The proteins are made up of three parts, one of which is G-alpha. When G-alpha receives a signal from its receptor, it becomes activated. To turn on, G-alpha needs to release a molecule called GDP – which is bound to G-alpha when turned off – and instead bind to another molecule called GTP. However, it remains unclear how exactly GDP is released when it receives a signal from its receptor.

Faulty G-alphas have been linked to many diseases, including cancer and heart conditions. However, current treatments do not currently target this part of G-protein signaling. To develop new drugs in the future, we first need a better understanding about the critical steps driving G-alpha activation, such as the release of GDP.

Now, Sun, Singh et al. used computer simulations and mathematical models to investigate how G-alpha is activated, and to identify the structural changes underlying the release of GDP. The simulations allow to observe how the atoms within G-alpha behave and were obtained from citizen-scientist volunteers, who ran simulations on their personal computers using the Folding@home app.

Together, they generated an enormous amount of data that would normally take over 150 years to collect with one computer. Subsequent analyses identified the critical atomic motions driving the release of GDP and a network of amino acids located within G-alpha. These amino acids allow G-alpha to act like a switch and connect the part that receives the signal from the receptor to the GDP-binding site. In the future, this model could serve as a platform for developing drugs that target G-alpha and shed more light into how signals are transmitted within our cells.

DOI: https://doi.org/10.7554/eLife.38465.002

(*Sunahara et al., 1997*; *Leipe et al., 2002*; *Westfield et al., 2011*). The high similarity of these structures and the binding of GDP or GTP deep in the protein's core suggests that activation occurs by adoption of other conformational states that allow GDP release (*Lambright et al., 1994*). One intermediate in G protein activation was suggested by the first crystal structure of a GPCR-bound G protein in which the Ras-like and helical domains of Gα are hinged apart and GDP has dissociated (*Rasmussen et al., 2011*). Structural analysis has led to the proposal of a universal mechanism for G protein activation (*Flock et al., 2015*). In this model, GPCR binding induces translation of H5 away from H1, which increases disorder in H1 and the P-loop (or Walker A motif [*Leipe et al., 2002*]) to facilitate GDP release. However, there is evidence that additional intermediates may be involved in Gα activation, (*Oldham and Hamm, 2008*; *Westfield et al., 2011*; *Liang et al., 2017*; *Hilger et al., 2018*) and the functional importance of this conformational ensemble has been previously suggested (*Furness et al., 2016*). Furthermore, mutagenesis and nuclear magnetic resonance (NMR) studies have suggested important roles for other structural elements (*Toyama et al., 2017*; *Sun et al., 2015*; *Goricanec et al., 2016*).

Molecular dynamics simulations promise to capture the entire mechanism of G protein activation and synthesize the wealth of experimental data on this process. Methodological advances now enable simulations to capture millisecond timescale processes for proteins with less than 100 residues (*Lindorff-Larsen et al., 2011*). For example, it is now possible to capture the binding or release of small molecules (*Buch et al., 2011*; *Bowman and Geissler, 2012*; *Silva et al., 2011*; *Plattner and Noé, 2015*; *Tiwary et al., 2015*) and peptides (*Plattner et al., 2017*; *Zhou et al., 2017*) from small proteins. Impressive simulations on the ANTON supercomputer have revealed critical conformational dynamics of G proteins in their inactive and active states, elucidating the role of domain opening in GDP unbinding (*Dror et al., 2015*; *Yao et al., 2016*). However, even this specialized hardware could

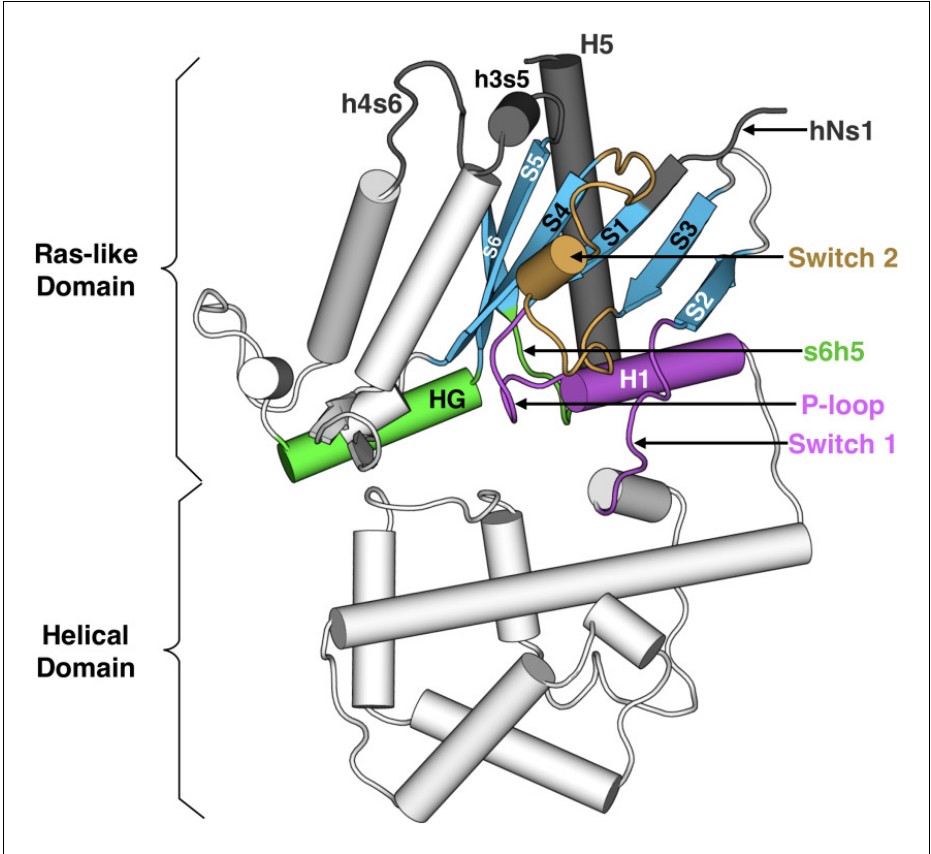

**Figure 1.** Structure of Gαq with key secondary structure elements labeled according to the Common Gα Numbering (CGN) system. The coloring scheme highlights the GPCR binding interface (gray), GDP phosphate-binding regions (pink), GDP nucleotide-binding regions (green), β-sheets (blue), and switch 2 (orange).
DOI: https://doi.org/10.7554/eLife.38465.003

not capture the entire process of G protein activation and GDP release due to the size of the Gα subunit (>300 residues) and the slow kinetics of GDP dissociation ($\sim 10^{-3}$ min$^{-1}$) (*Chidiac et al., 1999*; *Ross, 2008*; *Mukhopadhyay and Ross, 1999*).

Here, we introduce an approach to capture rare or long-timescale events, such as GDP release, and reveal the mechanism of Gα activation. As a test of this methodology, we apply it to Gαq, which has one of the slowest GDP release rates (*Chidiac et al., 1999*) and is frequently mutated in uveal melanoma (*Van Raamsdonk et al., 2009*; *Van Raamsdonk et al., 2010*). To highlight aspects of the activation mechanism that we propose are general to all G proteins, we focus our analysis on the behavior of secondary structure elements and amino acids that are conserved across Gα domains. Our approach first combines two powerful sampling methods, metadynamics (*Laio and Parrinello, 2002*) and Markov state models (MSMs), (*Bowman et al., 2014*) to observe GDP release and identify the rate-limiting step for this slow process. Then we use our recently developed CARDS method (*Singh and Bowman, 2017*), which quantifies correlations between both the structure and disorder of different regions of a protein, to identify the allosteric network connecting the GPCR- and nucleo-tide-binding sites. The resulting model is consistent with a wealth of experimental data and leads to a number of predictions, described below. Taken together, our results provide the most comprehensive model of G protein activation to date. Based on this success, we expect our approach to be valuable for studying other rare events, including conformational changes and unbinding processes.

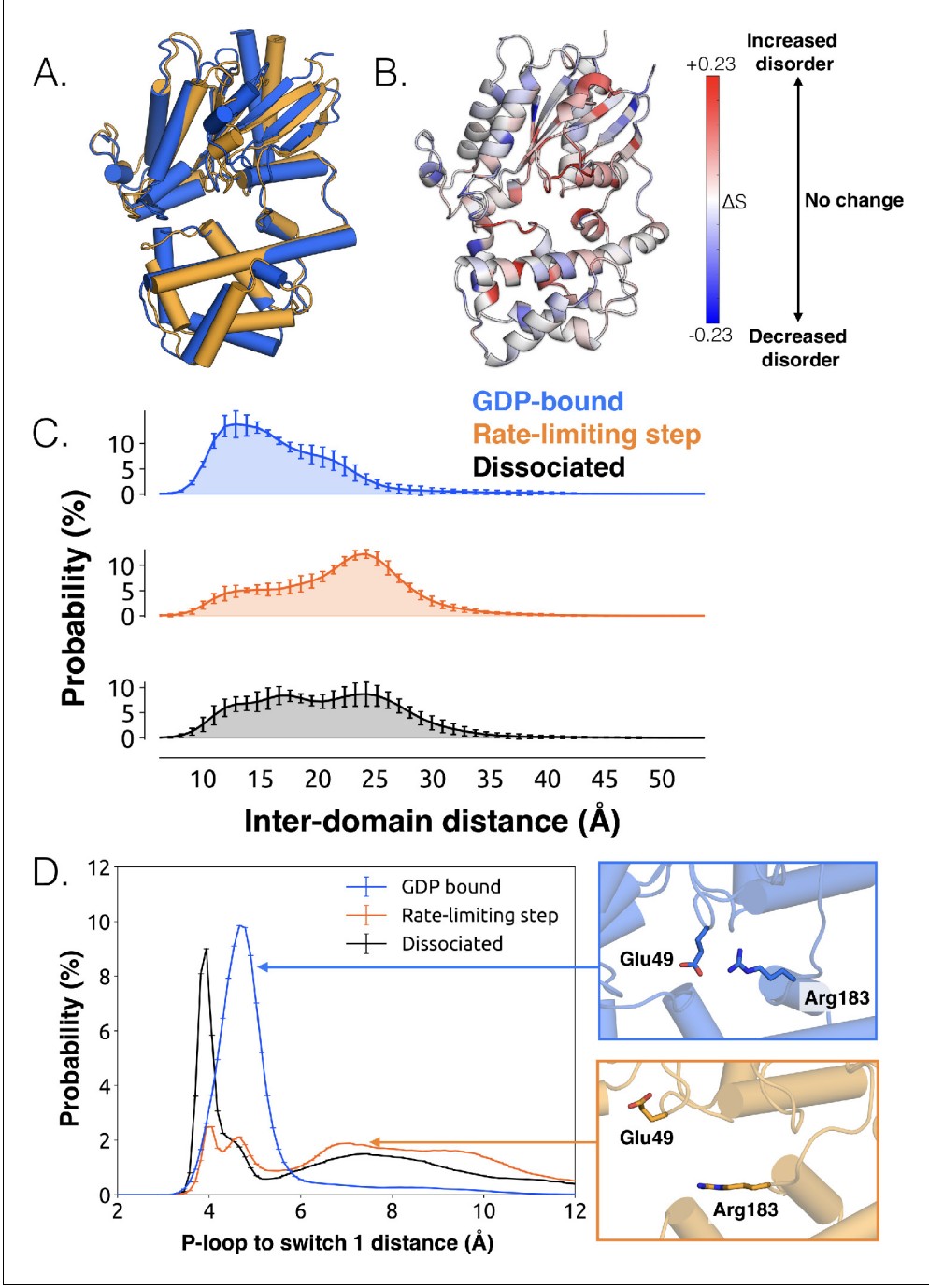

**Figure 2.** Structural and dynamical changes during the rate limiting step for GDP release. (**A**) Overlay of representative structures before (blue) and after (orange) the rate limiting step. (**B**) Change in conformational disorder (Shannon entropy) across the rate-limiting step, according to the color scale on the right. (**C**) Histograms of inter-domain distances before (top, blue) and after (middle, orange) the rate limiting step, along with the inter-domain distance distribution after GDP is released (bottom, black). Inter-domain distance is measured using residues analogous to those used in DEER experiments (**Dror et al., 2015**), Leu97[H.HA.29] in the helical domain and Glu250[G.H3.4] on H3. (**D**) Distribution of distances between Glu49[G.s1h1.4] and Arg183[G.hfs2.2] (left) before (blue) and after (orange) the rate-limiting step, as well as after GDP release (black). Representative structures of the interacting residues are also shown (right).

DOI: https://doi.org/10.7554/eLife.38465.004

The following figure supplements are available for figure 2:

*Figure 2 continued on next page*

*Figure 2 continued*

**Figure supplement 1.** Free-energy surface from metadynamics simulations of GDP release for the full Gαq (blue) and truncated form (green, without the last five C-terminal residues).
DOI: https://doi.org/10.7554/eLife.38465.005
**Figure supplement 2.** Overlay of representative structures of Gαq when bound to GDP (blue) or across the rate-limiting step (orange).
DOI: https://doi.org/10.7554/eLife.38465.006
**Figure supplement 3.** Changes in the structure (left) and disorder (right) of specific regions across the rate-limiting step.
DOI: https://doi.org/10.7554/eLife.38465.007
**Figure supplement 4.** Distribution of distances between the side-chains of K275[G.s5hg.1] and D155[H.hdhe.5] for the GDP-bound state (blue), across the rate-limiting step (orange), and upon GDP dissociation (black).
DOI: https://doi.org/10.7554/eLife.38465.008
**Figure supplement 5.** Implied timescales for the Markov state model.
DOI: https://doi.org/10.7554/eLife.38465.009

## Results and discussion

### Capturing G-protein activation and GDP release in atomic detail

We reasoned that studying the mechanism of spontaneous GDP release from a truncated form of Gαq would be representative of the mechanism of GPCR-mediated activation of the heterotrimeric G protein while minimizing the computational cost of our simulations. This hypothesis was inspired by previous work demonstrating that a protein's spontaneous fluctuations are representative of the conformational changes required for the protein to perform its function (*Boehr et al., 2006*; *Fraser et al., 2009*; *Changeux and Edelstein, 2011*). Therefore, we hypothesized that GPCRs stabilize conformational states that G proteins naturally, albeit infrequently, adopt in the absence of a receptor. We chose to focus on Gα since it forms substantial interactions with GPCRs, compared to the relatively negligible interactions between GPCRs and G protein β and γ subunits. This view is supported by the fact that GPCRs and 'mini' G proteins, essentially composed of just the Ras-like domain of Gα, recapitulate many features of GPCR-G protein interactions (*Carpenter et al., 2016*). We also reasoned that truncating the last five residues of Gαq would facilitate the activation process. These residues contact Gα in some GDP-bound structures but not in GPCR-bound structures, (*Lambright et al., 1996*; *Noel et al., 1993*) and removing these residues promotes GDP release due to a reduced GDP-binding affinity (*Denker et al., 1992*; *Marin et al., 2002*). Taken together, such evidence suggests that the last five residues of Gαq help stabilize the inactive state and that removing them would accelerate activation. In support of this hypothesis, we find that the energetic barrier to GDP release is lower in metadynamics simulations of the truncated variant than for full-length Gαq (*Figure 2—figure supplement 1*). These simulations, and those described hereafter, were initiated from an X-ray structure of the Gαq heterotrimer bound to GDP and an inhibitor of nucleotide exchange (*Nishimura et al., 2010*); Gβγ and the inhibitor were excluded from all simulations.

To observe G-protein activation, we developed a variant of adaptive seeding (*Huang et al., 2009*) capable of capturing slow processes like ligand unbinding that are beyond reach of conventional simulation methods. First, we use metadynamics (*Tiwary et al., 2015*; *Laio and Parrinello, 2002*; *Dama et al., 2014*) to facilitate broad sampling of conformational space by biasing simulations to sample conformations with different distances between GDP and Gαq. Doing so forces GDP release to occur but provides limited mechanistic information because adding a biasing force can distort the system's kinetics or cause the simulations to sample high-energy conformations that are not representative of behavior at thermal equilibrium. To overcome these limitations, we chose starting conformations along release pathways observed by metadynamics as starting points for standard molecular dynamics simulations, together yielding an aggregate simulation time of 122.6 μs. These simulations should quickly relax away from high-energy conformations and provide more accurate kinetics. Then we use these simulations to build an MSM (*Source data 1*). MSMs are adept at extracting both thermodynamic *and* kinetic information from many standard simulations that, together, cover larger regions of conformational space than any individual simulation

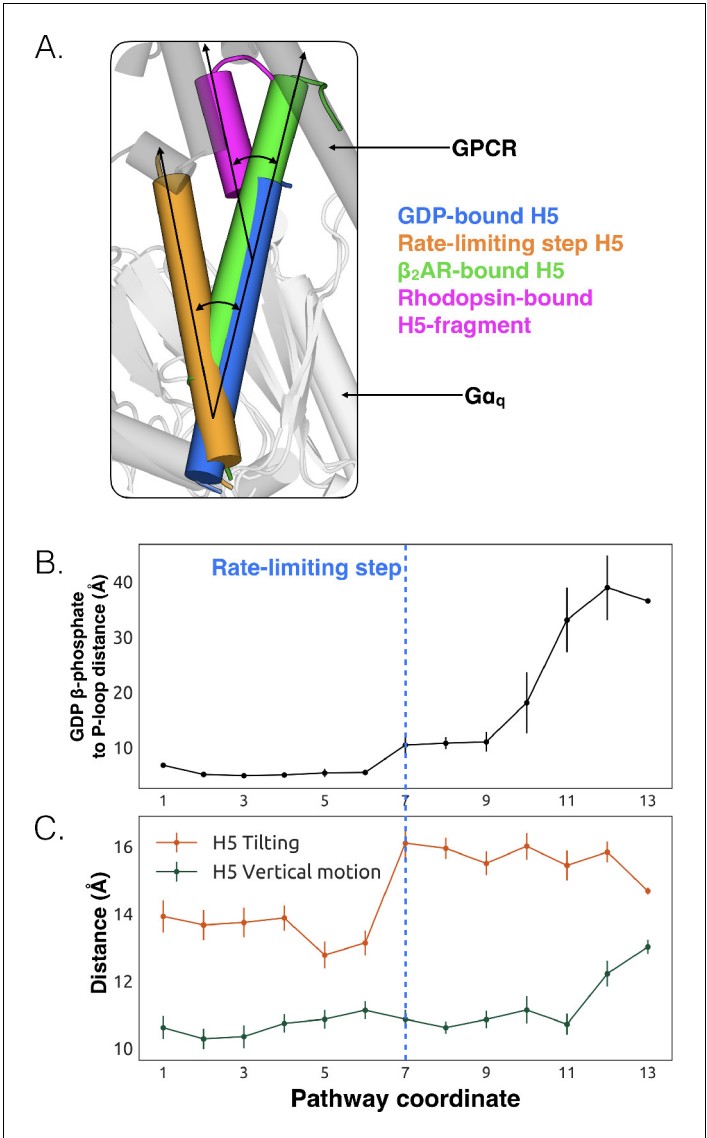

**Figure 3.** Tilting of H5 is correlated with GDP release but translation of H5 is not. (A) Displacements of H5 relative to the GDP-bound crystal structure (blue). The three other orientations of H5 come from the rate-limiting step in our model (orange), the co-crystal structure of $G\alpha s$ and β2AR (green, PDB ID 3SN6) (*Hilger et al., 2018*), and the co-crystal structure of a C-terminal fragment of $G\alpha t$ and rhodopsin (magenta, PDB ID 3PQR) (*Van Raamsdonk et al., 2010*). The black arrows highlight the change in orientation of the long axis of each helix. A representative GPCR (gray) and $G\alpha$ (white) structure are shown for reference. (B) GDP release distance across the highest flux pathway, defined as the distance from the GDP β-phosphate to the center of mass between residues Lys52[G.H1.1], Ser53[G.H1.2], and Thr54[G.H1.3] on H1. The state marking the rate-limiting step is highlighted by the blue dashed line. (C) H5 motion across the highest flux pathway. The distances measured here representing H5 motion are taken from the same states as in B. H5 tilting (orange) is measured by the distance between Leu349[G.H5.16] and Tyr325[G.S6.2]. Likewise, H5 vertical motion (green) is measured by the distance between Thr334[G.H5.1] and Phe341[G.H5.8]. The rate-limiting step is marked with the blue dashed line, extended down from B.

DOI: https://doi.org/10.7554/eLife.38465.010

The following source data and figure supplements are available for figure 3:

**Source data 1.** Table of measurements comparing tilting and translation of H5 across PDB structures and MD simulation.

DOI: https://doi.org/10.7554/eLife.38465.013

*Figure 3 continued on next page*

*Figure 3 continued*

**Figure supplement 1.** Probability distribution of the distance between Leu349[G.H5.16] on H5 and Phe194[G.S2.6] on S2 to monitor the tilting motion of H5 upon GDP release when bound to GDP (blue), across the rate-limiting step (orange), and upon GDP dissociation (black).
DOI: https://doi.org/10.7554/eLife.38465.011

**Figure supplement 2.** H5 vertical motion is sampled across GDP release simulations.
DOI: https://doi.org/10.7554/eLife.38465.012

(*Bowman et al., 2014*). Related approaches have successfully captured the dynamics of small model systems (*Biswas et al., 2018*) and RNA polymerase (*Zhang et al., 2016*).

This protocol enabled us to capture the entire mechanism of G-protein activation, including GDP release and the rate-limiting step for this process. Identifying the rate-limiting step for this process is of great value because GDP release is the rate-limiting step for G-protein activation and downstream signaling. Therefore, the key structural and dynamical changes responsible for activation should be apparent from the rate-limiting conformational transition for this dissociation process.

To identify the rate-limiting step, we applied transition path theory (*Noé et al., 2009*; *Weinan and Vanden-Eijnden, 2006*) to find the highest flux paths from bound structures resembling the GDP-bound crystal structure to fully dissociated conformations. Then, we identified the least probable steps along the 10 highest flux release pathways (*Figure 2A* and *Figure 2—figure supplement 2*), which represent the rate-limiting step of release. By comparing the structures before and after the rate-limiting step, we define the bound state as all conformations where the distance from the center of mass of GDP's phosphates to the center of mass of three residues on H1 that contact the GDP β-phosphate (Lys52[G.H1.1], Ser53[G.H1.2], and Thr54[G.H1.3]) is less than 8 Å. Consistent with this definition, this distance remains less than 8 Å throughout the entirety of 35.3 μs of GDP-bound simulations.

The conformational changes we observe during the rate-limiting step are consistent with previous structural and biochemical work. For example, we observe that the Ras-like and helical domains separate (*Figure 2C* and *Source data 3*), as observed in DEER experiments (*Van Eps et al., 2011*) and previous simulations (*Dror et al., 2015*). This finding is consistent with the intuition that these domains must separate to sterically permit GDP release, and that this separation is driven by the disruption of multiple inter-domain interactions. For example, we note a disrupted salt bridge between K275[G.s5hg.1] and D155[H.hdhe.5] (*Figure 2—figure supplement 4*), previously identified in structural studies (*Flock et al., 2015*). Domain opening is accompanied by disruption of a key salt bridge between Glu49[G.s1h1.4] of the P-loop and Arg183[G.hfs2.2] of switch 1 (*Figure 2D* and *Source data 3*), as well as an increase in the disorder of many of the surrounding residues (*Figure 2B* and *Figure 2—figure supplement 3A*), consistent with the proposal that this salt bridge stabilizes the closed, GDP-bound state (*Liang et al., 2017*).

While domain opening is necessary for GDP release, previous simulations suggest it is insufficient for unbinding (*Dror et al., 2015*). Indeed, we also see that this opening is necessary but not sufficient for GDP unbinding, as the Ras-like and helical domains often separate prior to release (*Figure 2C* and *Source data 3*). Notably, the Ras-like and helical domains only separate by ~10 Å during the rate-limiting step. In contrast, these domains separate by 56 Å in the first structure of a GPCR-G-protein complex. This result suggests that GDP release may have occurred long before a G protein adopts the sort of widely opened conformations observed in crystallographic structures (*Rasmussen et al., 2011*).

## Tilting of H5 helps induce GDP release

We also observe less expected conformational changes associated with GDP release. The most striking is tilting of H5 by about 26° (*Figure 3A*, and *Figure 3—figure supplement 1*). We find that H5 tilting correlates strongly with the distance between GDP and Gαq along the highest flux dissociation pathway (*Figure 3B* and *Source data 1*). In particular, substantial tilting of H5 is coincident with the rate-limiting step for GDP release. This tilting contrasts with X-ray structures and the universal mechanism, in which H5 is proposed to translate along its helical axis towards the GPCR, initiating the process of GDP release (*Figure 3A*). During our simulations we also observe translation of H5, but it is not correlated with the rate-limiting step of GDP release (*Figure 3C*, *Figure 3—figure*

*supplement 2*, and *Source data 3*). Therefore, we are not merely missing an important role for translation due to insufficient sampling.

The potential importance of H5 tilting is supported by other structural data. For example, a crystal structure of rhodopsin (*Choe et al., 2011*) with a C-terminal fragment from H5 of Gαt shows a similar degree of tilting (*Figure 3A*). Also, the tilt of H5 varies in crystal structures of the β2AR-Gs complex (*Rasmussen et al., 2011*), two different GLP-1 receptor-Gs complexes (*Liang et al., 2017*; *Zhang et al., 2017*), and an A2AR-mini-Gs complex (*Carpenter et al., 2016*). The potential relevance of tilting has also been acknowledged by a number of recent works (*Flock et al., 2015*; *Rasmussen et al., 2011*; *Oldham et al., 2006*) including four recently published structures of receptor-G-protein complexes across which H5 also shows a broad range of tilting and translational motion (*Koehl et al., 2018*; *Draper-Joyce et al., 2018*; *García-Nafría et al., 2018*; *Kang et al., 2018*). Interestingly, the tilting and translation we observe falls within the observed range of tilting and translational motions that H5 undergoes in available GPCR-G protein complex structures (*Rasmussen et al., 2011*; *Koehl et al., 2018*; *Draper-Joyce et al., 2018*; *García-Nafría et al., 2018*; *Kang et al., 2018*), providing support that conformational selection plays an important role (*Figure 3—source data 1*). Finally, H5 is translated toward the GPCR in the A2AR-mini-Gs structure but GDP remains bound (*Carpenter et al., 2016*). The authors of that study originally suggested that one of the mutations in mini-Gs decouples H5 translation from GDP release. However, given that we see GDP release without H5 translation in our simulations, it is also possible that amino acid substitutions required to create mini-Gs instead mitigate H5 tilting. Both of these models are consistent with the fact that some of the mutations in mini-Gs stabilize the GDP-bound state (*Sun et al., 2015*).

We propose that tilting of H5 is an early step in the GDP release process, which is followed by upward translation of this helix to form a GPCR-G protein complex primed to bind GTP. This hypothesis stems from our observation that tilting of H5 is coincident with the rate-limiting step for GDP release, while translation of H5 only becomes stable after GDP dissociates (*Figure 3C*). This model is consistent with previous suggestions that G-protein activation occurs through a series of sequential interactions with a GPCR (*Oldham and Hamm, 2008*; *Rasmussen et al., 2011*). Another possibility is that any displacement of H5, whether tilting or translation, may be sufficient to trigger GDP release.

## Identification of the allosteric network that triggers GDP release

While conformational changes of H5 are important for Gα activation, other regions could also play a role (*Hilger et al., 2018*; *Sun et al., 2015*). However, it is not straightforward to determine what other structural elements contribute to activation or their importance relative to H5. Our hypothesis that spontaneous motions of a protein encode functionally relevant conformational changes suggests that the coupling between the GPCR- and nucleotide-binding sites of Gα should be present in simulations of the inactive protein; This provides a means to identify key elements of this allosteric network. To test this hypothesis, we ran 35.3 μs of simulation of GDP-Gαq. Then we applied the CARDS method (*Singh and Bowman, 2017*) to detect correlations between both the structure and dynamical states of every pair of dihedral angles. Structural states are determined by assigning dihedral angles to the three dominant rotameric states for side-chains (gauche+, gauche-, and trans) and the two dominant rotameric states for backbone dihedrals (cis and trans). Dynamical states are determined by whether a dihedral angle remains in a single rotameric state (ordered) or rapidly transitions between multiple rotameric states (disordered). These pairwise correlations serve as a basis for quantifying the correlation of every residue to a target site, such as the GPCR-binding site. Combining these correlations with structural and dynamical changes in our model of GDP release provides a basis for inferring how perturbations to the GPCR-binding site are transmitted to the nucleotide-binding site. We focus our analysis on the most direct routes for communication between the GPCR- and nucleotide-binding sites by following correlated motions that emanate from structural elements that directly contact GPCRs until they reach the GDP-binding site. There are correlations between many other elements of Gαq, including parts of the helical domain, that branch off of this allosteric network. Such correlations may be important for aspects of Gα function besides activation, but are beyond the scope of the present study, which focuses on how GPCR-binding impacts nucleotide release.

To understand how H5 tilting triggers GDP release, we first identified structural elements with strong coupling to H5 and then worked our way outward in repeated iterations until we reached the

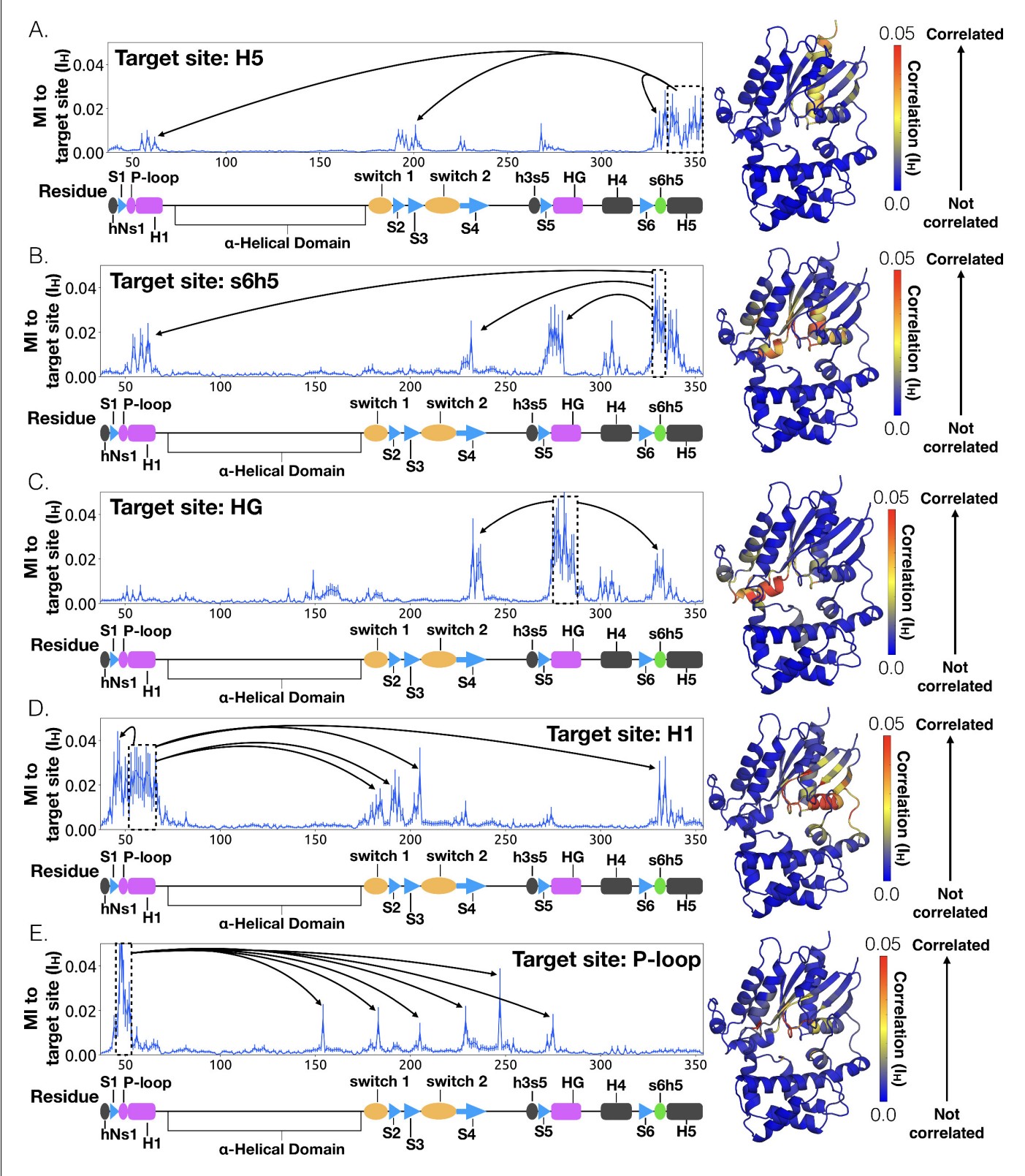

**Figure 4.** Allosteric network connecting H5 motion to the nucleotide binding-site via s6h5. CARDS data showing communication per residue to a target site (dashed box) is plotted (left) and mapped onto the structure of Gαq (right) for (**A**) H5, (**B**) s6h5, (**C**) HG, (**D**) H1, and (**E**) the P-loop. Arrows indicate regions of importance with significant communication to the target site.
DOI: https://doi.org/10.7554/eLife.38465.014

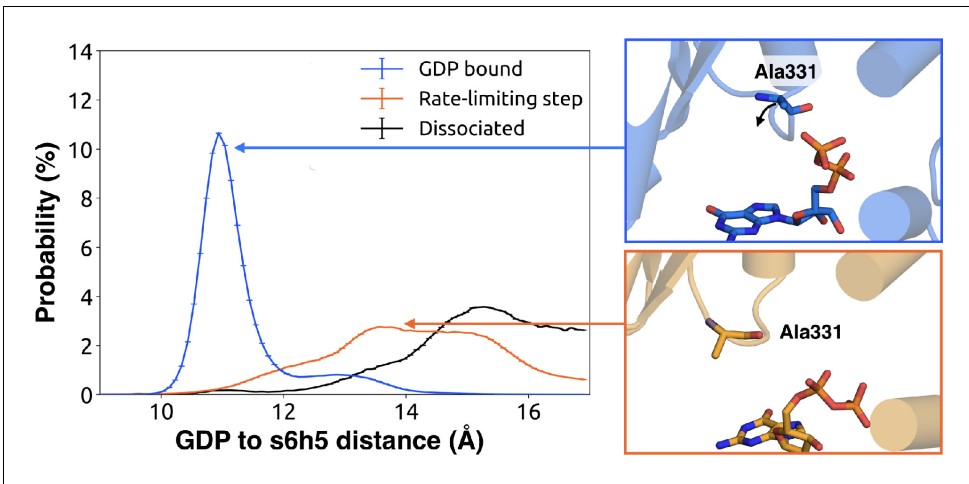

**Figure 5.** Change in the s6h5 loop conformation across the rate-limiting step. Distribution of distances (left) from GDP's β-phosphate to Ala331[G.s6h5.3] on the s6h5 loop before (blue) and after (orange) the rate-limiting step, as well as after GDP release (black). Representative structures of the s6h5 loop (right) are shown for before (top right, blue) and after (bottom right, orange) the rate limiting step.

DOI: https://doi.org/10.7554/eLife.38465.015

nucleotide-binding site (*Figure 4* and *Source data 2*). This analysis reveals that tilting of H5 directly communicates with and impacts the conformational preferences of the s6h5 loop, which contacts the nucleobase of GDP (*Figure 4*, *Figure 5*, *Source data 2*, and *Source data 3*). During the rate-limiting step, these contacts are broken and there is increased disorder in the s6h5 loop, particularly Ala331 of the TCAT motif (*Figure 5* and *Figure 2—figure supplement 3B*). The importance of the TCAT motif in our model is consistent with its conservation and the fact that mutating it accelerates GDP release (*Iiri et al., 1994*; *Posner et al., 1998*; *Thomas et al., 1993*). Based on our model, we propose these mutations accelerate release by weakening shape complementarity with GDP.

We also observe an important role for communication from H5 to H1, consistent with the universal mechanism. In particular, H1 is strongly coupled with the s6h5 loop (*Figure 4B* and *Source data 2*), which is sensitive to displacement of H5. In the rate-limiting step, s6h5 moves away from H1, contributing to an increase in disorder of H1 and the P-loop (*Figure 2—figure supplement 3A* and *Figure 2—figure supplement 3B*). Increased disorder in a set of residues that directly contact the GDP phosphates (Glu49[G.s1h1.4], Ser50[G.s1h1.5], Gly51[G.s1h1.6], Lys52[G.H1.1], and Ser53[G.H1.2]) likely contributes to a reduced affinity for GDP (*Figure 2—figure supplement 3A*). Increased disorder in these residues also contributes to disruption of the salt bridge between Glu49[G.s1h1.4] of the Ras-like domain and Arg183[G.hfs2.2] of the helical domain, facilitating inter-domain separation.

We further note that the s6h5 loop impacts the nucleotide-binding site through allosteric coupling with the HG helix, which also contacts GDP via Lys275[G.s5hg.1] and Asp277[G.HG.2] (*Figures 4E* and *6*). The disorder of both of these residues increases during the rate-limiting step, consistent with observations of increased mobility in HG upon receptor-mediated activation (*Oldham and Hamm, 2008*). There are also correlations between the P-loop and Lys275[G.s5hg.1] on Helix G (*Figure 4E* and *Source data 2*), which result from the disruption of a key salt bridge between Lys275[G.s5hg.1] and Glu49[G.s1h1.4] on the P-loop during the rate-limiting step (*Figure 6* and *Source data 3*). Lys275[G.s5hg.1] is conserved across all Gα families, suggesting it plays an important role in the stability or function of the protein. However, attempts to experimentally examine the role of this residue by mutating Lys275[G.s5hg.1] have resulted in aggregation (*Sun et al., 2015*). Our simulations suggest Lys275[G.s5hg.1] plays an important role in stabilizing the GDP-bound state and that breaking the salt bridge with Glu49[G.s1h1.4] facilitates GDP release. This finding demonstrates the utility of our atomistic simulations, as we can examine the role of residues that are difficult to probe experimentally.

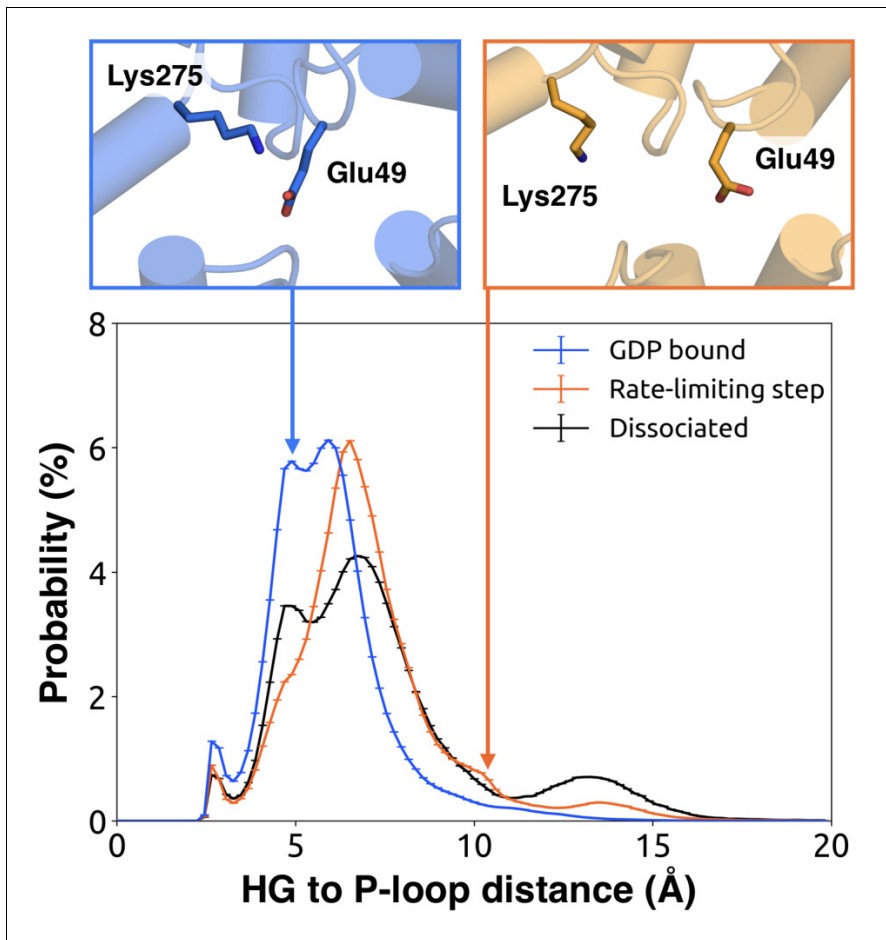

**Figure 6.** Probability distributions of the distance between the side-chains of Lys275[G.s5hg.1] and Glu49[G.s1h1.4]. Distributions were computed for the bound (blue), rate-limiting step (orange), and dissociated (black) states. Representative structures (above) for the bound (left, blue) and rate-limiting step (right, orange) are included with residues as sticks.

DOI: https://doi.org/10.7554/eLife.38465.016

## H1 and the β-sheets are communication hubs

To identify other important structural elements in the allosteric network underlying G protein activation, we followed correlated motions emanating from other sites that are known to interact directly with GPCRs, including the hNs1 loop, the h3s5 loop, and the h4s6 loop (*Oldham and Hamm, 2008*). We find that h3s5 and h4s6 are largely isolated, suggesting they play a role in forming a stable GPCR-G protein complex but not in the allosteric mechanism that triggers GDP release. This finding is consistent with sequence analysis suggesting these structural elements are important determinants of the specificity of GPCR-Gα interactions (*Flock et al., 2017*). In contrast, the hNs1 loop has strong correlations with β-strands S1-S3 (*Figure 7* and *Source data 2*). These strands, in turn, communicate with H1, switch 1, and the P-loop.

Integrating our correlation analysis with structural insight from the rate-limiting step described above suggests an important role for S1-S3 in a complex allosteric network that couples the GPCR- and nucleotide-binding sites (*Figures 7* and *8*, *Figure 7—figure supplement 1*, and *Source data 2*). S2 and S3 twist relative to S1 and away from H1 (*Figures 2A* and *9*, *Figure 9—figure supplement 1*, and *Source data 3*). This twisting disrupts stacking between Phe194[G.S2.6] on S2 and His63[G.H1.12] on H1 and increases disorder of side-chains in H1 (*Figures 2B* and *9*, *Figure 2—figure supplement 3C*, and *Source data 3*). Increased disorder in H1 is also a crucial component of the proposed universal mechanism, but in that model translation of H5 is the key trigger for changes in H1. The role

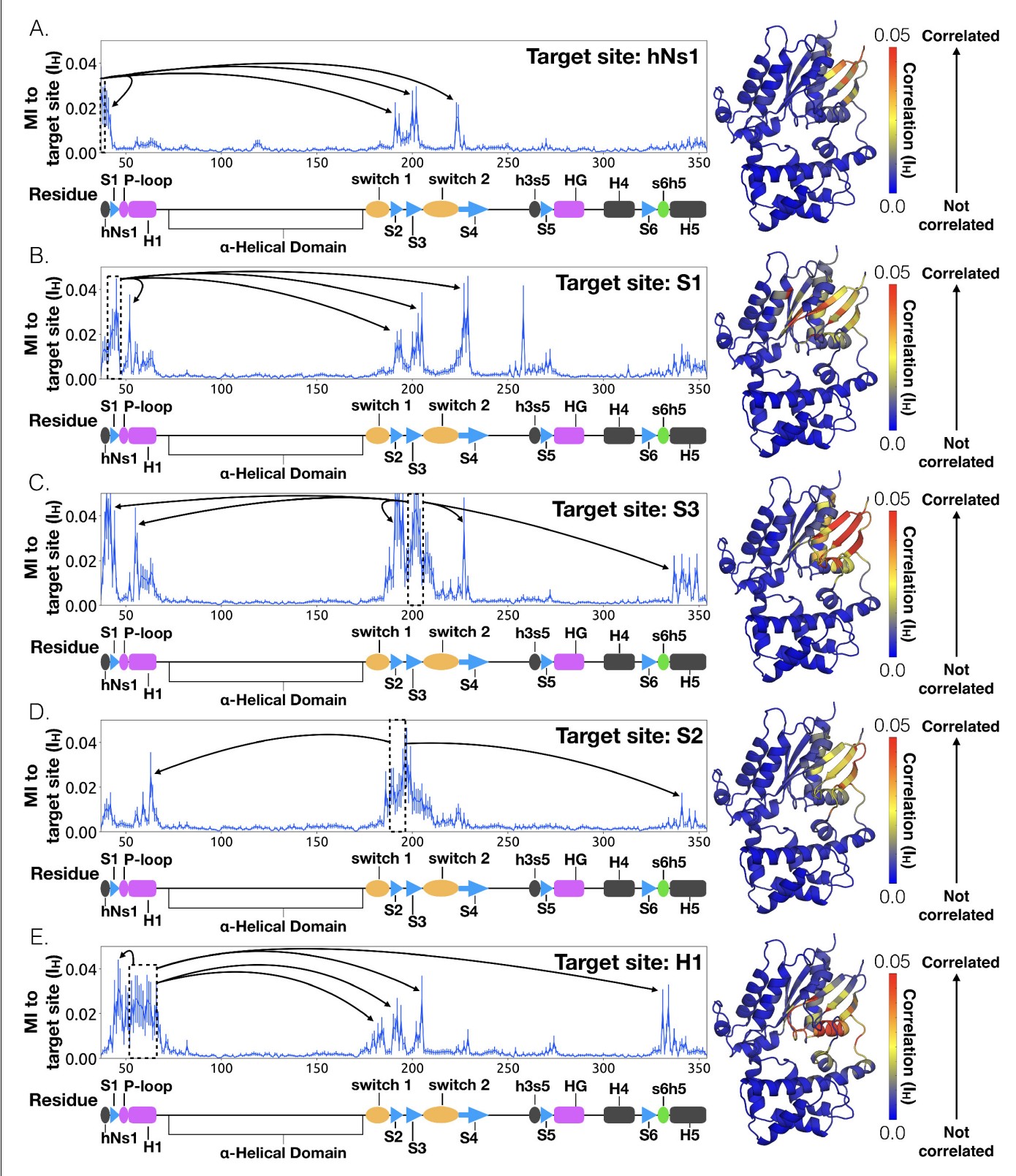

**Figure 7.** Allosteric network connecting hNs1 motion to the nucleotide-binding site via the β-sheets. CARDS data showing communication per residue to a target site (dashed box) is plotted (left) and mapped onto the structure of Gαq (right) for (**A**) hNs1, (**B**) S1, (**C**) S3, (**D**) S2, and (**E**) H1. Arrows indicate regions of importance with significant communication to the target site.

DOI: https://doi.org/10.7554/eLife.38465.017

*Figure 7 continued on next page*

*Figure 7 continued*

The following figure supplement is available for figure 7:

**Figure supplement 1.** Allosteric network connecting hNs1 contacts to the P-loop and switch 1 via S4.

DOI: https://doi.org/10.7554/eLife.38465.018

for the β-sheets in our model is consistent with previous work identifying interactions between S2 and H1 (*Flock et al., 2015*), NMR experiments showing chemical exchange in the methyls of S1-S3 upon receptor binding (*Toyama et al., 2017*), and mutational data. In particular, Flock et al. have previously noted the important interaction between residues Phe194[G.S2.6] and His63[G.H1.12] (*Flock et al., 2015*).

The importance of H1 and β-strands S1-S3 is underscored by mapping the global communication of every residue onto a structure of Gα (*Figure 8—figure supplement 1*). The global communication of a residue is the sum of its correlations to every other residue and is a useful metric for identifying residues that are important players in allosteric networks (*Singh and Bowman, 2017*). Interestingly, these β-strands and H1 have higher global communication than H5 and the s6h5 loop. This suggests that H1 and the β-sheets integrate conformational information from multiple sources, including the hNs1 loop, and not just H5. The importance of the β sheets and H1 for allosteric communication is consistent with their conservation (*Sun et al., 2015*), which may not simply reflect the role they play in protein folding and stability, as had been suggested previously (*Sun et al., 2015*; *Hatley et al., 2003*).

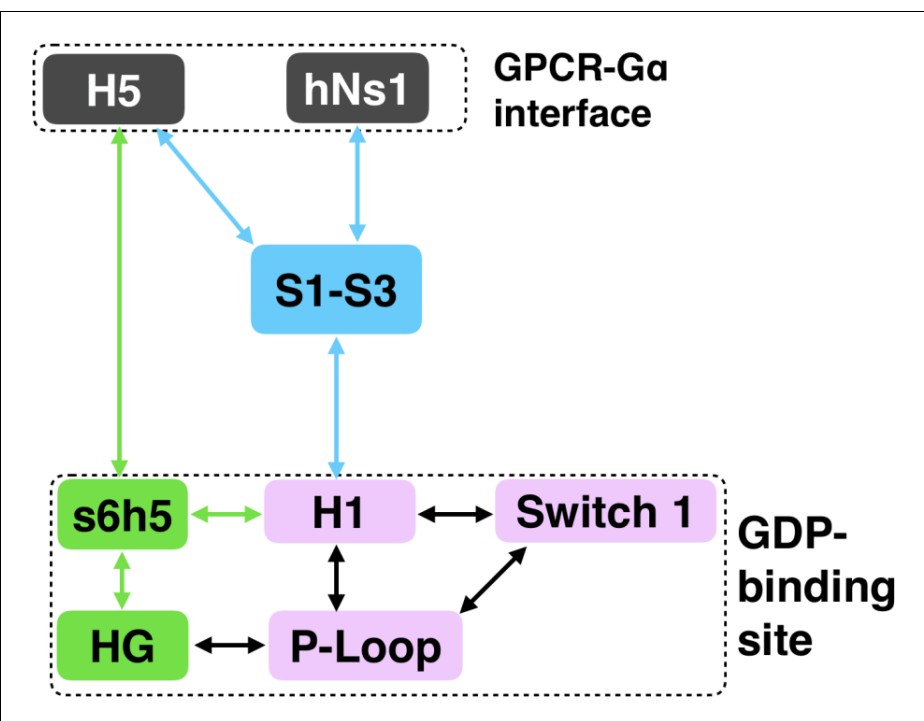

**Figure 8.** Allosteric network connecting the GPCR- and nucleotide-binding interfaces. The coloring scheme corresponds to that used in *Figure 1*, highlighting the GPCR binding interface (gray), GDP phosphate-binding regions (pink), GDP nucleotide-binding regions (green), and the β-sheets (blue).

DOI: https://doi.org/10.7554/eLife.38465.019

The following figure supplement is available for figure 8:

**Figure supplement 1.** Global communication of each residue in the Ras-like domain mapped onto the structure of Gαq, colored based on the scale (right).

DOI: https://doi.org/10.7554/eLife.38465.020

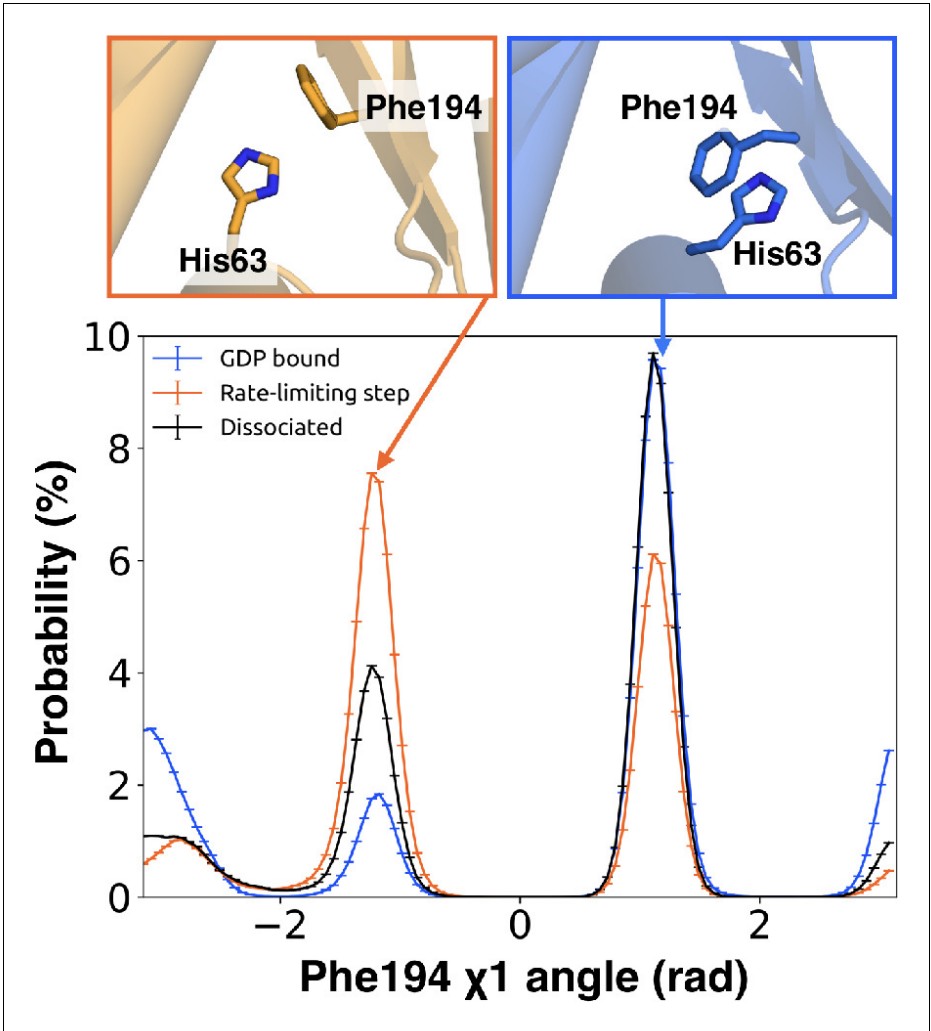

**Figure 9.** π–stacking between S2 and H1 is disrupted during the rate-limiting step. Distribution of the χ1 angle (bottom) of Phe194[G.S2.6] on S2 before (blue) and after (orange) the rate-limiting step, as well as after GDP release (black). Representative structures of Phe194[G.S2.6] and His63[G.H1.12] (top) corresponding to before and after the rate-limiting step are also shown.

DOI: https://doi.org/10.7554/eLife.38465.021

The following figure supplement is available for figure 9:

**Figure supplement 1.** Probability distributions of the twist angle between S1 and S3.

DOI: https://doi.org/10.7554/eLife.38465.022

## GDP release alters the structure and dynamics of the Gβ-binding site

We also find that switch 2 has strong correlations with the nucleotide-binding site, especially switch 1 (*Figure 7—figure supplement 1* and *Source data 2*). Given that switch 2 is a major component of the interface between Gα and Gβ, this communication could enable GDP release to trigger dissociation of Gα from Gβγ. Examining the rate-limiting step for GDP release reveals that switch 2 shifts towards the nucleotide-binding site (*Figure 10* and *Source data 3*) and exhibits increased conformational disorder (*Figure 2B* and *Figure 2—figure supplement 3D*). These findings are consistent with previous kinetic studies postulating that switch 2 dynamics are impacted prior to GDP release (*Herrmann et al., 2004*).

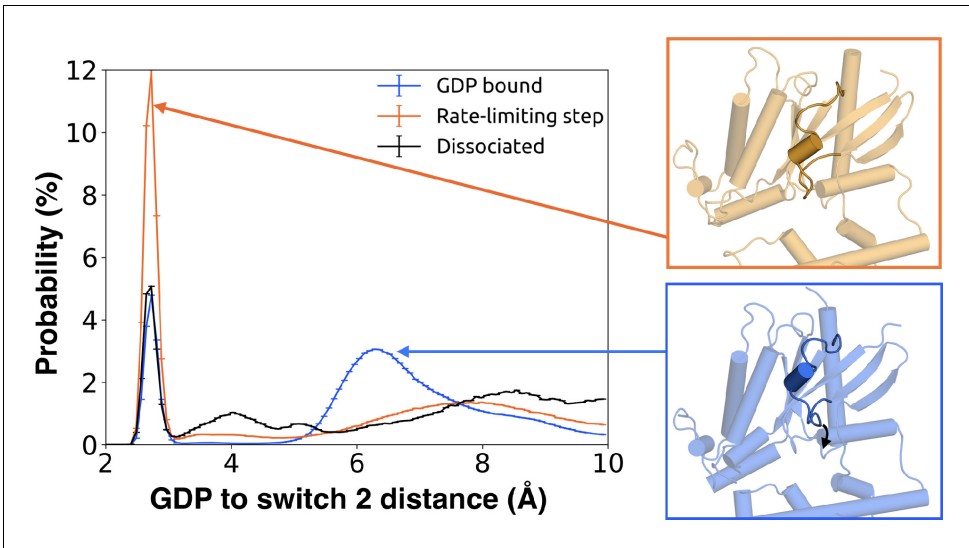

**Figure 10.** Switch 2 moves towards GDP across the rate-limiting step. Distance distribution (left) of Gly207[G.s3h2.1] to GDP before (blue) and after (orange) the rate-limiting step, as well as after GDP release (black). Representative structures of Switch 2 motion are shown (right).

DOI: https://doi.org/10.7554/eLife.38465.023

## Conclusions

We have succeeded in simulating G protein activation, including both the allosteric coupling between the GPCR- and nucleotide-binding sites of Gαq and consequent unbinding of GDP. Our results reveal a previously unobserved intermediate that defines the rate-limiting step for GDP release and, ultimately, G protein activation. Our model synthesizes a wealth of experimental data and previous analyses. For example, we identify an important role for coupling from H5 to the s6h5 loop and H1 that is consistent with a previously proposed universal mechanism for G protein activation. However, we also find that this allosteric network incorporates the hNs1 loop, β-strands S1-S3, and the HG helix. Strands S1-S3 and H1 serve as hubs in this network, simultaneously integrating information from both H5 and the hNs1 loop. Our observation is consistent with previous postulates that information flows from H5 and hNs1 to H1 (*Preininger et al., 2013*). It is important to note that our model was extracted using simulations of Gαq, and so some correlations or changes in conformation and dynamics may apply only to Gαq. However, by focusing our analysis on secondary structure elements and residues that are shared across all Gα homologs, our model likely captures a universal 'skeleton' of changes involved in Gα activation, expanding upon a previously proposed universal mechanism for Gα activation (*Flock et al., 2015*). The consistency of our model with a wide variety of structural and biochemical data suggests that it is a promising foundation for future efforts to understand the determinants of GPCR-Gα interaction specificity, how mutations cause aberrant signaling and disease, and how small molecule inhibitors modulate Gα activation. Our model also adds weight to the growing appreciation for the fact that a protein's spontaneous fluctuations encode considerable information about its functional dynamics (*Koehl et al., 2018*; *Draper-Joyce et al., 2018*; *García-Nafría et al., 2018*; *Kang et al., 2018*). Construction of our model was enabled by a powerful combination of simulation methods, namely metadynamics and MSMs. In the future, we expect this methodology will prove valuable for understanding other slow conformational changes and unbinding processes.

# Materials and methods

## Molecular dynamics simulations of GDP unbinding

### System preparation

We used the crystal structure of Gα from the heterotrimeric Gαq protein (PDB entry 3AH8) as the initial structure to set up our simulations (*Nishimura et al., 2010*). The first 36 residues were removed prior to simulation, as they come from Gαi, along with the Gβ and Gγ subunits which were removed to minimize the system size and maximize our chances of observing GDP release (*Oldham and Hamm, 2008*).

The protein structure was solvated in a dodecahedron box of TIP3P water (*Jorgensen et al., 1983*) that extended 1 nm beyond the protein in every dimension. A single $Mg^{2+}$ ion was added coordinating the phosphate groups of GDP as its presence accelerates GDP release (*Zhang et al., 2000*). Thereafter, $Na^+$ and $Cl^-$ were added to produce a neutral system at 0.15 M NaCl. The final system consists of one GDP, one Gαq, 57 $Cl^{-1}$, 64 $Na^{+1}$, one $Mg^{2+}$, and 18696 TIP3P water molecules, for a total of 61,893 atoms.

Molecular dynamics (MD) simulations were carried out using Gromacs (*Abraham et al., 2015*; *Van Der Spoel et al., 2005*) and the AMBER03 force field (*Duan et al., 2003*). The force field parameters of GDP were obtained from the AMBER Parameter Database (http://research.bmh.manchester.ac.uk/bryce/amber) (*Meagher et al., 2003*). The system was energy minimized with the steepest descent algorithm for 50,000 steps until the maximum force fell below 100 kJ/mol/nm using a step size of 0.02 nm and a cut-off distance of 0.9 nm for the neighbor list, electrostatic interactions and van der Waals interactions. Afterward, the solvent was relaxed by a NVT simulation of 100 ps with the constraint of 1000 kJ/mol/nm applied to the protein heavy atoms and 2 fs as the time step. In this relaxation simulation, the temperature of the system is constrained to 300 K using V-rescale thermostat (with a time constant of 0.1 ps) (*Bussi et al., 2007*). A cut-off distance of 1 nm was used for the van der Waals, short-range electrostatic interactions. Periodic boundary conditions are applied to x, y and z directions. The Particle-Mesh-Ewald method is employed to recover the long-range electrostatic interactions with 0.16 nm as the grid spacing and with a fourth order spline (*Kolafa and Perram, 1992*). All the bonds connecting to hydrogens are constrained using LINCS algorithm (*Hess, 2008*). After the NVT relaxation, the system further underwent an NPT simulation for one ns with the time step of 2 fs for equilibration. The simulation parameters here are kept the same as the NVT relaxation except for the application of Parrinello-Rahman barostat for pressure coupling (*Parrinello and Rahman, 1981*). After these equilibration runs, the constraint on heavy atoms were removed for all subsequent production runs. Virtual sites were used to allow for a 4 fs time-step (*Feenstra et al., 1999*). We then applied a three-step protocol to simulate GDP release.

### Step one: MD simulations of the GDP-bound state

We performed 100 parallel simulations of the GDP bound state on the Folding@home (*Shirts and Pande, 2000*) distributed computing environment for an aggregate simulation time of 35.3 μs.

### Step two: Metadynamics simulations

We subsequently ran metadynamics simulations (*Laio and Parrinello, 2002*; *Tribello et al., 2014*) initiated from conformations generated in step one to actively promote GDP release. Starting conformations were selected from step one by clustering protein conformations into 625 states using a hybrid K-center/K-medoids method (*Beauchamp et al., 2011*) with a 2 Å cutoff. The 10 most populated states included conformations with large inter-domain separation distances between the Ras-like and AH domains as measured by the angle formed between the α-carbon atoms of Leu97[H.HA.29], Asn82[H.HA.14], Ile258[G.H3.12], and Glu250[G.H3.4]. This inter-domain angle ranged from −30° to 0°. From these states, three representative structures were chosen with inter-domain angles of −6° (Conf. 1),−10° (Conf. 2), and −26.6° (Conf. 3) as starting conformations for metadynamics simulations, which were run on PLUMED (*Tribello et al., 2014*). We defined two collective variables for our metadynamics simulations: 1) the distance between GDP's phosphate groups and the backbone of Lys52[G.H1.1]-Thr54[G.H1.3] in Gαq subunit (CV1), and 2) the RMSD of GDP to the starting conformation (CV2).

In metadynamics, a history-dependent biased potential $V_G(S,t)$ is added to the two selected CVs.

$$V_G(S,t) = \int_0^t dt' \, \omega \, exp\left(-\sum_{i=1}^{d} \frac{\left(S_i(R) - S_i(R(t'))\right)^2}{2\sigma_i^2}\right)$$

where t represents time, S are collective variables, $\omega$ is the energy rate and $\sigma_i$ controls the width of the Gaussian for the ith collective variable. Summing up the Gaussians allows us to obtain the biased potential $V_G$. The free energy $-F(S)$ is derived by the assumption,

$$\lim_{t\to\infty} V_G(S,t) \sim -F(S)$$

The metadynamics simulations were repeated three times for each selected representative structure using different Gaussian widths (*Table 1*). We set the Gaussian height to 1.5 kJ/mol.

We observed the release of GDP from Gα in the metadynamics simulations and obtained the free-energy landscape for the release. We then applied the string method (*Maragliano et al., 2006*) to detect potential release pathways and use these conformations as the seeds for simulations in step 3. We can use the function $\chi(\alpha)$ to represent $\chi(C_1, C_2)$ which showing the minimum free-energy path. We thus have

$$\frac{dZ(\alpha)}{d\alpha} = \sum_{k=1}^{n} \frac{\partial \theta_i(\chi(\alpha))}{\partial \chi_k} \frac{d\chi_k}{d\alpha}$$

which is parallel to

$$\sum_{k=1}^{n} \frac{\partial \theta(\chi(\alpha))}{\partial \chi_k} \frac{dF(\chi_k)}{d\chi_k} = \sum_{j,k=1}^{n} \frac{\partial \theta_i(\chi(\alpha))}{\partial \chi_k} \frac{\partial \theta_j(\chi(\alpha))}{\partial \chi_k} \frac{dF(z(\alpha))}{dZ(\alpha)}$$

The average of the tensor

$$\sum_{j,k=1}^{n} \frac{\partial \theta_i(\chi(\alpha))}{\partial \chi_k} \frac{\partial \theta_j(\chi(\alpha))}{\partial \chi_k}$$

can be represented as

$$M_{ij}(z) = \Omega^{-1} e^{\beta F(z)} \int_{R^N} \sum_{k=1}^{n} \frac{\partial \theta_i(\chi(\alpha))}{\partial \chi_k} \frac{\partial \theta_j(\chi(\alpha))}{\partial \chi_k} e^{-\beta F(z)} \prod_{v=1}^{N} \delta(z_v - \theta_v(\chi)) d\chi$$

The points determining the minimum free-energy path along the surface satisfy $0 = \left(M_{ij}(z)\Delta F(z(\alpha))\right)^{\perp}$.

**Table 1.** Details of metadynamics simulations.

| Starting conformation | Width of CV1 (Å) | Width of CV2 (Å) | Number of conformations selected |
|---|---|---|---|
| 1 | 0.1 | 0.1 | 171 |
| 1 | 0.08 | 0.03 | 132 |
| 1 | 0.03 | 0.01 | 504 |
| 2 | 0.1 | 0.1 | 145 |
| 2 | 0.08 | 0.03 | 141 |
| 2 | 0.03 | 0.01 | 320 |
| 3 | 0.1 | 0.1 | 198 |
| 3 | 0.08 | 0.03 | 186 |
| 3 | 0.03 | 0.01 | 288 |

DOI: https://doi.org/10.7554/eLife.38465.024

We applied this method (*Maragliano and Vanden-Eijnden, 2007*; *Maragliano et al., 2014*) to obtain a minimum free energy path, extracting 2085 conformations along potential GDP release pathways. Notably, we only observed the transition from the bound state and the unbound state for one time in a single metadynamics simulation. This implies that the predicted free-energy surface from metadynamics cannot be used to describe the release events accurately. In spite of this, we can still use the conformations along the release pathway explored by metadynamics to seed unbiased parallel MD simulations.

## Step three: Metadynamics-seeded MD simulation of GDP release

Lastly, we carried out unbiased MD simulations initiated from the 2085 conformations obtained from metadynamics using the Folding@home platform (*Shirts and Pande, 2000*). A total of 122.6 μs of simulation was generated in this step. All the following analyses are based on unbiased MD simulations.

### Identifying the allosteric network with CARDS

To determine how the GPCR- and GDP-binding regions communicate with one another, we applied the CARDS (*Singh and Bowman, 2017*) methodology to simulations of the GDP-bound state of Gαq. CARDS measures communication between every pair of dihedrals via both correlated changes in structural motions and dynamical behavior. Structural states are captured by discretizing backbone $\Phi$ and $\psi$ dihedrals into two structural states (*cis* and *trans*), while side-chain $\chi$ angles are placed into three states (*gauche+*, *gauche-*, and *trans*). Every dihedral is also parsed into dynamical states, capturing whether the dihedral is stable in a single state (ordered), or rapidly transitioning between multiple states (disordered). These dynamical states are identified using two kinetic signatures of dihedral motion: the average time a dihedral persists in a structural state (an ordered timescale), and the typical timescale for transitions between structural states (a disordered timescale). Parsing into dynamical states utilizes a two-step process by (i) calculating the distribution or ordered and disordered times from the simulations and (ii) assigning each period of time between two consecutive transitions into ordered and disordered states based on which distribution the time between two transitions is most consistent with.

From these states, a holistic communication ($I_H(X, Y)$) is computed for every pair of dihedrals $X$ and $Y$:

$$I_H(X, Y) = \overline{I_{ss}(X, Y)} + \overline{I_{sd}(X, Y)} + \overline{I_{ds}(X, Y)} + \overline{I_{dd}(X, Y)}$$

where $\overline{I_{ss}(X, Y)}$ is the normalized mutual information between the structure (i.e., rotameric state) of dihedral $X$ and the structure of dihedral $Y$, $\overline{I_{sd}(X, Y)}$ is the normalized mutual information between the structure of dihedral $X$ and the dynamical state of dihedral $Y$, $\overline{I_{ds}(X, Y)}$ is the normalized mutual information between the dynamical state of dihedral $X$ and the structure of dihedral $Y$, and $\overline{I_{dd}(X, Y)}$ is the normalized mutual information between the dynamical state of dihedral $X$ and the dynamical state of dihedral $Y$. The Mutual Information ($I$) is

$$I(X, Y) = \sum_{x \in X} \sum_{y \in Y} p(x, y) \log\left(\frac{p(x, y)}{p(x)p(y)}\right)$$

where $x \in X$ refers to the set of possible states that dihedral $X$ can adopt, $p(x)$ is the probability that dihedral $X$ adopts state $x$, and $p(x, y)$ is the joint probability that dihedral $X$ adopts state $x$ and dihedral $Y$ adopts state $y$. Normalized mutual information is computed using the maximum possible mutual information for any specific mode of communication.

From the pairwise correlation for every dihedral-pair, we extracted how much each individual residue communicates with a target site of interest via bootstrapping with 10 random samples with replacement. After locating the group of residues communicating most strongly with a specific target site, we set this newly identified group as the new target site; The iteration of this process allows us to identify a pathway of communication from one region of interest to another. Here, we set the GPCR contact sites as our initial target sites. We then iteratively used this approach to identify pathways connecting these contact sites with the GDP-binding site of Gαq.

## Markov state model construction

We clustered Gαq conformations and GDP binding states separately and combined the assignments to build a Markov State Model using MSMbuilder (*Beauchamp et al., 2011*; *Bowman et al., 2009*) and enspara (*Porter et al., 2018*). First, we clustered protein conformations into 5040 states using a hybrid k-center/k-medoids method with 1.8 Å cutoff. Then we clustered the GDP-binding state into 321 states using the automatic partitioning algorithm (APM) (*Sheong et al., 2015*) with a residence time of 2 ns. By combining the assignments from protein conformations and the GDP-binding states, we obtained a total of 221,965 states. The implied timescales of this MSM show Markovian behavior with a lag time of 5 ns (*Swope et al., 2004*). (*Figure 2—figure supplement 5*). Analyses of distances, angles, and dihedrals of interest were carried out using bootstrapping with ten random samples, with replacement. Results were insensitive to varying the number of bootstrapped samples between 5 and 30. Histograms were generated using 100 bins.

## Quantifying conformational disorder

The disorder of every residue was measured by computing Shannon entropy (*Shannon, 1948*) of each dihedral as they are natural degrees of freedom for describing protein dynamics. Shannon entropy ($H$) is defined as

$$H(X) = -\sum_{x \in X} p(x) \log(p(x))$$

where $x \in X$ refers to the set of possible states that dihedral $X$ can adopt and $p(x)$ is the probability that dihedral $X$ adopts state $x$. Dihedral angles were calculated using MDTraj (*McGibbon et al., 2015*) and assigned to discrete rotameric states as described above using CARDS. The entropy of a single residue was computed by summing up the entropies of its dihedrals, and normalized by the residue's maximum possible Shannon entropy. This maximum possible Shannon entropy, using a flat distribution for the appropriate number of bins, is referred to as the 'channel capacity' and has been used to normalize other information-theoretic metrics (*Singh and Bowman, 2017*). Summing entropies within a residue establishes an upper bound on the degree of motion for a single residue, while ignoring intra-residue correlations between dihedrals.

## Identification of the rate-limiting step for GDP release

We used transition path theory (TPT) (*Noé et al., 2009*; *Weinan and Vanden-Eijnden, 2006*) to find the highest flux paths from the bound state to the unbound state (*Metzner et al., 2009*). The bound state was defined as all clusters that satisfied two criteria: (i) GDP is within 6 Å of the backbone atoms of Lys52-Thr54, and (ii) GDP has an RMSD <0.5 Å to its crystallographic conformation. The unbound state was defined as all clusters with GDP > 55 Å from the binding pocket. The rate-limiting step was identified by finding the bottleneck in the highest flux paths. To obtain this, we first calculate the flux between states i and j along any possible unbinding path using

$$f_{ij}^{BU} = \begin{cases} \pi_i q_i^- T_{ij} q_j^+ & i \neq j \\ 0 & i = j \end{cases}$$

where $q_i^+$ is the committor probability from the bound to the unbound state, and $q_i^-$ is $1 - q_i^+$; $\pi_i$ is the weighted probability, and $T_{ij}$ is the transition matrix. The highest flux paths can be identified by maximizing the fluxes between the bound states and the unbound states using

$$c(w) = \min\left(f_{i_l i_{l+1}}^+ | l = 1..n_w - 1\right)$$

where $i_l$ are intermediate states. From this, the slowest step was extracted as the minimum flux step of the highest flux release pathway.

## Acknowledgements

We thank TE Frederick and TD Todd for their helpful discussion and insight. We are grateful to the Folding@home users for computing resources. This work was funded by National Institutes of Health grants R01GM12400701, R01GM044592 and R01GM12409301, as well as by the National Science

Foundation CAREER Award MCB-1552471. GRB holds a Career Award at the Scientific Interface from the Burroughs Wellcome Fund and a Packard Fellowship for Science and Engineering from The David and Lucile Packard Foundation.

## Additional information

### Funding

| Funder | Grant reference number | Author |
| --- | --- | --- |
| National Institutes of Health | Grant R01GM12400701 | Gregory R Bowman |
| National Science Foundation | CAREER Award MCB-1552471 | Gregory R Bowman |
| Burroughs Wellcome Fund | Career Award at the Scientific Interface | Gregory R Bowman |
| David and Lucile Packard Foundation | Packard Fellowship for Science and Engineering | Gregory R Bowman |
| National Institutes of Health | Grant R01GM044592 | Kendall J Blumer Gregory R Bowman |
| National Institutes of Health | Grant R01GM12409301 | Kendall Blumer Gregory R Bowman |

The funders had no role in study design, data collection and interpretation, or the decision to submit the work for publication.

### Author contributions

Xianqiang Sun, Conceptualization, Data curation, Software, Formal analysis, Investigation, Visualization, Methodology, Writing—review and editing; Sukrit Singh, Data curation, Software, Formal analysis, Validation, Investigation, Visualization, Methodology, Writing—original draft, Writing—review and editing; Kendall J Blumer, Conceptualization, Resources, Data curation, Formal analysis, Supervision, Funding acquisition, Validation, Investigation, Visualization, Project administration, Writing—review and editing; Gregory R Bowman, Conceptualization, Resources, Data curation, Formal analysis, Supervision, Funding acquisition, Investigation, Visualization, Methodology, Writing—original draft, Project administration, Writing—review and editing

### Author ORCIDs

Sukrit Singh https://orcid.org/0000-0003-1914-4955
Gregory R Bowman http://orcid.org/0000-0002-2083-4892

### Decision letter and Author response

Decision letter https://doi.org/10.7554/eLife.38465.030
Author response https://doi.org/10.7554/eLife.38465.031

## Additional files

### Supplementary files

• Source data 1. Zipped archive containing the Markov state model, including a transition probability matrix, a transition counts matrix, and state populations. Indices are self-consistent between files – that is, the population of state 1 corresponds to row 1 in the transition probability matrix. This model was used to generate *Figure 2*, *3*, *5*, *6*, *9* and *10*.
DOI: https://doi.org/10.7554/eLife.38465.025

• Source data 2. Compressed archive containing CARDS calculations of communication from each residue to a target site. Each file is stored in a comma-separated-value (CSV) format and the file-name contains the target site of the calculation. These data were used to generate *Figure 4*, *7* and *8*.
DOI: https://doi.org/10.7554/eLife.38465.026

• Source data 3. Numerical data used to generate histograms in this work, stored in a comma-separated-value (CSV) format and compressed into a zipped archive. Each file contains 3 columns representing 1) the order parameter measured, 2) probabilities and 3) errors as measured by bootstrapping. Files are separated into sub-folders representing the GDP-bound state, after the rate-limiting step, and upon GDP dissociation. The data in these folders were used to generate *Figure 2*, *3*, *5*, *6*, *9* and *10*.
DOI: https://doi.org/10.7554/eLife.38465.027

• Transparent reporting form
DOI: https://doi.org/10.7554/eLife.38465.028

## Data availability

Summary data for all figures are made available with this manuscript. Specifically, MSM data, CARDS data, and numerical data for histograms are each provided as zipped archives. Simulation data are available upon request as there is no standard repository for such data, especially given the size of our dataset (387 GB).

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
