## [Decision Letter]

Thank you for submitting your article "Simulation of spontaneous G protein activation reveals a new intermediate driving GDP unbinding" for consideration by *eLife*. Your article has been favorably reviewed by two peer reviewers, and the evaluation has been overseen by José Faraldo-Gómez as the Reviewing Editor and John Kuriyan as the Senior Editor. Both reviewers have agreed to reveal their identity: Alan Grossfield and Madan Babu.

The reviewers have discussed the reviews with one another and the Reviewing Editor has drafted this decision to help you prepare a revised submission.

Summary:

Sun et al. from the Bowman group report the findings from the simulation of spontaneous G protein activation. Specifically, they provide detailed molecular insights into the process by which G protein activation drives GDP release and highlight that tilting of helix 5 has an important role in the kinetics of GDP release. The study is based upon sophisticated computational methods and seeks to understand an important biological problem. The manuscript is well-written and the data are presented clearly.

Essential revisions:

1) Earlier studies have demonstrated the influence the specific ligand-receptor complex and receptor conformation on G protein dynamics and kinetics (e.g. Furness et al., 2016). Those earlier findings ought to be discussed and contrasted with the authors' proposal. The authors' approach appears to be based, by design, on a conformational selection perspective of the G protein. It is however a concern whether simulations of an isolated G protein can capture the range of conformational states accessible in the context of the receptor. It would be informative to analyze the analogous tilting/twisting angles in recently published structures of receptor-G protein complexes.

2) It is key that the manuscript is revised to make clear that the observations made are for Gαq (from the Abstract onwards), and that it remains to be shown whether they are applicable to other G proteins. The authors are however encouraged to discuss whether they anticipate their findings to be universal, and their rationale. On a related note, the authors are asked to revise the manuscript as needed to ensure a fair characterization of previous proposals. For example, Flock et al. (2015) discuss a mechanism they see as common but also incomplete, thus requiring other elements that might vary among G proteins. Specifically: "While the conserved residue contacts are crucial for Gα activation, non-conserved positions can still be important for allosteric activation in distinct Gα proteins. […] Thus, the conserved universal mechanism probably represents the 'skeleton' that can be incorporated into different contexts in different Gα proteins to maintain a conserved mechanism of allosteric activation and yet permit specific binding to the receptor".

3) On a technical note, how was the per-residue maximum Shannon entropy estimated? Is it just the entropy for evenly distributing probability across all bins? And what is the justification for simply summing entropies? That would be correct if each individual dihedral were independent of all others, but that is not the case (that information precisely is used elsewhere). Please provide an estimate for the error introduced by this approximation (or an appropriate reference if this issue has been discussed elsewhere).

---

## [Author Response]

Essential revisions:1) Earlier studies have demonstrated the influence the specific ligand-receptor complex and receptor conformation on G protein dynamics and kinetics (e.g. Furness et al., 2016). Those earlier findings ought to be discussed and contrasted with the authors' proposal. The authors' approach appears to be based, by design, on a conformational selection perspective of the G protein. It is however a concern whether simulations of an isolated G protein can capture the range of conformational states accessible in the context of the receptor. It would be informative to analyze the analogous tilting/twisting angles in recently published structures of receptor-G protein complexes.

We agree that our approach assumes conformational selection, which may be a limitation in some cases, and that comparing to available structural data on receptor-G protein complexes can help alleviate these concerns. Comparing the conformational heterogeneity in our models to that in recently published structures of receptor-G protein complexes is a reasonable way to address this concern.

Characterizing the available crystal structures using the same order parameters we used for tilting and translation in our simulations on analogous residues (Figure 3—source data 1) reveals a broad range of possible motion. Our simulations’ tilting and translational measurements fall within these observed ranges, supporting our hypothesis that the fluctuations of isolated G proteins are similar to those that occur in the receptor-G protein complex (i.e. conformational selection).

A sentence referencing Figure 3—source data 1 and the structures used has been added:

“The potential relevance of tilting has also been acknowledged by a number of recent works (Flock et al., 2015; Rasmussen et al., 2011; Oldham et al., 2006) including four recently published structures of receptor-G protein complexes across which H5 also shows a broad range of tilting and translational motion (Koehl et al., 2018; Draper-Joyce et al., 2018; Garcίa-Nafrίa et al., 2018; Kang et al., 2018). Interestingly, the tilting and translation we observe falls within the observed range of tilting and translational motions that H5 undergoes in available GPCR-G protein complex structures, (Rasmussen et al., 2011; Koehl et al., 2018; Draper-Joyce et al., 2018; Garcίa-Nafrίa et al., 2018; Kang et al., 2018) providing support that conformational selection plays an important role (Figure 3—source data 1).”

We also thank the reviewers for pointing us to Furness et al., as it provides support for the role of conformational selection in our studies. Furness et al. suggest that distinct G protein conformations occur in response to agonists with low vs. high efficacy at the calcitonin receptor, and that these distinct conformational states might control GTP loading and subsequent turnover of receptor-G protein complexes. While their work focuses primarily on the GTP binding process, their data suggests that the ensemble of conformations a G protein adopts can have significant functional implications; a postulate that could equally apply to the GDP unbinding process. Therefore, this work reinforces the importance of understanding conformational heterogeneity within Gα to understand function. To that end, we have added a citation in the following sentence:

“However, there is evidence that additional intermediates may be involved in Gα activation (Oldham and Hamm, 2008; Westfield et al., 2011; Liang et al., 2017; Hilger, Masureel and Kobilka, 2007), and the functional importance of this conformational ensemble has been previously suggested (Furness et al., 2016).”

2) It is key that the manuscript is revised to make clear that the observations made are for Gαq (from the Abstract onwards), and that it remains to be shown whether they are applicable to other G proteins. The authors are however encouraged to discuss whether they anticipate their findings to be universal, and their rationale. On a related note, the authors are asked to revise the manuscript as needed to ensure a fair characterization of previous proposals. For example, Flock et al. (2015) discuss a mechanism they see as common but also incomplete, thus requiring other elements that might vary among G proteins. Specifically: "While the conserved residue contacts are crucial for Gα activation, non-conserved positions can still be important for allosteric activation in distinct Gα proteins. […] Thus, the conserved universal mechanism probably represents the 'skeleton' that can be incorporated into different contexts in different Gα proteins to maintain a conserved mechanism of allosteric activation and yet permit specific binding to the receptor".

We agree it is important to acknowledge that our data is for Gαq and which aspects of our findings are specific to this system, and which we propose are general to all G proteins. We propose that the level at which we have analyzed the simulation data likely describes a universal “skeleton” of conformational events, because (1) much of our analysis is at the secondary structure level and (2) the results we present for specific amino acids focus on conserved positions. To clarify this point, we have added the following to the Introduction:

“As a test of this methodology, we apply it to Gαq, which has one of the slowest GDP release rates (Chidiac, Markin and Ross, 1999) and is frequently mutated in uveal melanoma (Van Raamsdonk et al., 2009; Van Raamsdonk et al., 2010). To highlight aspects of the activation mechanism that we propose are general to all G proteins, we focus our analysis on the behavior of secondary structure elements and amino acids that are conserved across Gα domains.”

In the Conclusions section, we have also added the following:

“It is important to note that our model was extracted using simulations of Gαq, and so some correlations or changes in conformation and dynamics may apply only to Gαq. However, by focusing our analysis on secondary structure elements and residues that are shared across all Gα homologs, our model likely captures a universal ‘skeleton’ of changes involved in Gα activation, expanding upon a previously proposed universal mechanism for Gα activation (Flock et al., 2015).”

3) On a technical note, how was the per-residue maximum Shannon entropy estimated? Is it just the entropy for evenly distributing probability across all bins? And what is the justification for simply summing entropies? That would be correct if each individual dihedral were independent of all others, but that is not the case (that information precisely is used elsewhere). Please provide an estimate for the error introduced by this approximation (or an appropriate reference if this issue has been discussed elsewhere).

This is correct. We normalize the Shannon entropy using a flat distribution for the appropriate number of bins. The maximum possible entropy is therefore dictated by our binning, and represents the ‘channel capacity’ of a dihedral’s entropy. Summing entropies per residue provides an upper bound on the degree of motion, ignoring correlations within a residue between individual dihedrals. This is discussed further in Singh and Bowman (2017).

To appropriately note these points, we have added a sentence in the Materials and methods section:

“This maximum possible Shannon entropy, using a flat distribution for the appropriate number of bins, is referred to as the “channel capacity” and has been used to normalize other information-theoretic metrics (Singh and Bowman, 2017). Summing entropies within a residue establishes an upper bound on the degree of motion for a single residue, while ignoring intra-residue correlations between dihedrals.”